

# Genomic evaluation of the probiotic and pathogenic features of *Enterococcus faecalis* from human breast milk and comparison with the isolates from animal milk and clinical specimens

Lobna Badr[1,3], Muhammad Yasir[2,3], Areej A. Alkhaldy[4], Samah A. Soliman[5], Magdah Ganash[1], Safaa A. Turkistani[6], Asif A. Jiman-Fatani[7], Ibrahim A. Al-Zahrani[2,3] and Esam I. Azhar[2,3]

[1] Department of Biology, Faculty of Science, King Abdulaziz University, Jeddah, Saudi Arabia
[2] Department of Medical Laboratory Sciences, Faculty of Applied Medical Sciences, King Abdulaziz University, Jeddah, Saudi Arabia
[3] Special Infectious Agents Unit, King Fahd Medical Research Center, King Abdulaziz University, Jeddah, Saudi Arabia
[4] Department of Clinical Nutrition, Faculty of Applied Medical Sciences, King Abdulaziz University, Jeddah, Saudi Arabia
[5] Department of Nursing, Dr. Soliman Fakeeh Hospital, Jeddah, Saudi Arabia
[6] Medical Laboratory Sciences, Fakeeh College for Medical Sciences, Jeddah, Saudi Arabia
[7] Department of Clinical Microbiology and Immunology, Faculty of Medicine, King Abdulaziz University, Jeddah, Saudi Arabia

Corresponding author
Muhammad Yasir,
yasirkhattak.mrl@gmail.com

## ABSTRACT

*Enterococcus faecalis* is considered a probiotic, commensal lactic acid bacterium in human breast milk (HBM), but there are circulating antibiotic resistant and virulence determinants that could pose a risk in some strains. The study aimed to conduct genomic analysis of *E. faecalis* isolates recovered from HBM and animal milk and to evaluate their probiotic and pathogenic features through comparative genomics with isolates from clinical specimens (*e.g.*, urine, wound, and blood). Genomic analysis of 61 isolates was performed, including *E. faecalis* isolates recovered from HBM in Saudi Arabia. Genome sequencing was conducted using the MiSeq system. The fewest antibiotic resistance genes (*lsaA*, *tetM*, *ermB*) were identified in isolates from HBM and animal milk compared to clinical isolates. Several known and unknown mutations in the *gyrA* and *parC* genes were observed in clinical isolates. However, 11 virulence genes were commonly found in more than 95% of isolates, and 13 virulence genes were consistently present in the HBM isolates. Phylogenetically, the HBM isolates from China clustered with the probiotic reference strain Symbioflor 1, but all isolates from HBM and animal milk clustered separately from the clinical reference strain V583. Subsystem functions 188 of 263 were common in all analyzed genome assemblies. Regardless of the source of isolation, genes associated with carbohydrate metabolism, fatty acid, and vitamin biosynthesis were commonly found in *E. faecalis* isolates. In conclusion, comparative genomic analysis can help distinguish the probiotic or pathogenic potential of *E. faecalis* based on genomic features.

## INTRODUCTION

The rich fluid of human breast milk (HBM) containing essential nutrients, antimicrobial peptides, bioactive compounds, immunoglobulins, proteins, lactoferrin, carbohydrates, and immune cells is crucial for neonatal health (*Yi & Kim, 2021*). Traditionally, it was considered a sterile fluid, but recent studies have reported common microbial presence (streptococci, lactobacilli, bifidobacteria, and enterococci) in HBM (*Anjum et al., 2022*; *Huang et al., 2019*; *Jost et al., 2013*). These commensal milk bacteria are important for establishing infant gut microbiota, and probiotic traits of HBM microbes have been recently revealed in various studies (*Anjum et al., 2022*; *Fernandez et al., 2013*; *Le Doare et al., 2018*; *Reis et al., 2016*). The probiotic potential of HBM lactobacilli strains is well established; however, the literature regarding the probiotic potential of enterococcal strains is limited (*Anjum et al., 2022*; *Reis et al., 2016*).

Lactic acid bacteria (LAB) of the genus *Enterococcus* naturally colonize the genital tract, lower GI tract, and oral cavity of the human body (*Krawczyk et al., 2021*; *Murray, 1990*). The human intestine harbors *E. faecium* ($10^4$–$10^5$ cfu/g feces) and *E. faecalis* ($10^5$–$10^7$ cfu/g feces) (*Murray, 1990*). The enterococcal species either follow the entero-mammary pathway to translocate to the mammary glands from the maternal gut or the maternal skin (*Fernandez et al., 2013*; *Rodriguez, 2014*). These first neonatal GIT-colonizing LAB facilitate further development of the infant microbiome (*Dominguez-Bello et al., 2010*; *Fanaro et al., 2003*). *Enterococcus* spp. are also known to exert beneficial properties such as antimicrobial, cholesterol-lowering, regulation of the immune system, maintenance of normal intestinal microflora, antitumor, and antioxidant activities (*Al Atya et al., 2015*; *Anjum et al., 2022*; *Krawczyk et al., 2021*). Therefore, they are often used as probiotics to promote animal and human health and treat diseases (diarrhea, and irritable bowel syndrome) (*Franz et al., 2011*). They participate in nutrient metabolism (carbohydrates, proteins, and lipids), synthesize metabolites and vitamins, and maintain the pH of their niche (*Bagci et al., 2019*; *Panthee et al., 2021*). *Anjum et al. (2022)* have recently reported the isolation of *E. faecalis* isolates from HBM, which lacked most virulence factors and were significantly tolerant to bile, acid, and GIT digestive enzymes. They also displayed antibacterial activity against various pathogens (*Anjum et al., 2022*).

Despite the positive aspects, *E. faecalis* could act as opportunistic pathogens to cause nosocomial infections such as bacteremia, urinary tract infections, diarrhea, endocarditis, surgical site, and bloodstream infections (*Farman et al., 2019*; *Krawczyk et al., 2021*). Therefore, the safety verification of enterococcal probiotic strains requires a vigilant approach. The combination of virulence and multiple antibiotic resistance factors complicate their simple safety assessments and thus require comprehensive safety evaluations (*Baccouri et al., 2019*; *Dapkevicius et al., 2021*; *Krawczyk et al., 2021*). For example, milk-isolated *Enterococcus* spp. from healthy women had previously been shown to exhibit resistance to different clinical antibiotics, including tetracycline, gentamicin,

chloramphenicol, streptomycin, clindamycin, and quinupristin/dalfopristin (*Jimenez et al., 2013*; *Kozak et al., 2015*; *Landete et al., 2018*). Previous studies revealed the presence of antibiotic resistance and virulence genes in human milk-isolated *E. faecium* and *E. faecalis*, which could lead to the development of antibiotic resistance in offspring (*Jimenez et al., 2013*; *Kozak et al., 2015*; *Landete et al., 2018*). The species like *Staphylococcus aureus*, *E. faecium*, *E. faecalis*, and *Staphylococcus hominis* exhibit resistance to multiple antibiotics isolated from milk of healthy mothers (*Chen, Tseng & Huang, 2016*; *Huang et al., 2019*; *Saeed et al., 2023*). Enterococci carry different classes of virulence factors, including (a) externally secreted (serine protease, cytolysin, and gelatinase), (b) surface proteins (extracellular surface protein Esp, Acm/Ace adhesins, and Ebp pili), and (c) other virulence factors (hyaluronidase) (*Gilmore et al., 2002*).

The pathogenic and probiotic strains within the same species indicate genomic variation, which could lead to differential phenotypic features (*Panthee et al., 2021*). Enterococcal genomes have been widely studied in recent years, focusing on the comparative genomic analysis of clinical isolates. Multiple studies have elaborated on the safety aspects of non-clinical and clinical *E. faecalis* and *E. faecium* strains (*Hanchi et al., 2018*; *Kim & Marco, 2014*; *Krawczyk et al., 2021*; *Panthee et al., 2021*). However, literature regarding genomic analysis of HBM-isolated *E. faecalis* strains is limited. During the current study, a comparative genomic analysis was performed to identify unique and common genetic features among 61 *E. faecalis* strains isolated from animal milk, HBM, and clinical specimens. Whole genomes alignment was performed to establish phylogenetic relationships between isolated strains. Moreover, genomes were analyzed for secondary metabolite gene clusters, mobile genomic elements (MGE), virulence genes, and phages. Given the importance of antibiotic resistance in probiotics, antibiotic-resistance genes were also investigated.

## MATERIALS & METHODS

### Sampling and culture-depending screening

In this study, HBM samples were collected from three healthy mothers between 7 and 10 days' post full-term pregnancy in Saudi Arabia. These samples were called HBM_SA samples. The average age of mothers was 31 $\pm$ 2.6 years. The milk samples were collected under the sterile conditions described previously (*Hunt et al., 2011*). Ethical approval was obtained from the Institutional Review Board of the Dr. Soliman Fakeeh Hospital, Jeddah, under the number 77/IRB/2020. Four culture media of sheep blood agar, R2A agar, MacConkey agar, and MRS agar supplemented with 50% pasteurized milk were used to isolate bacteria from the HBM_SA samples. A serial dilution approach in peptone water was adopted, and 100 µl of the sample was spread on each culture medium. The colony-forming unit (CFU) was calculated after 48 h of incubation at 37 °C. The colonies were purified by sub-culturing and preserved in autoclaved 15% glycerol and 1% skim milk in distilled water at −80 °C.

The purified colonies were identified by MALDI-TOF based VITEK-MS (BioMérieux, France) system described previously (*Yasir et al., 2022*). The VITEK-MS run was validated

using the reference strain *Escherichia coli* ATCC 25922, following the manufacturer's standard protocol (BioMérieux, Marcy-l'Étoile, France). Antimicrobial susceptibility and minimum inhibitory concentration (MIC) of *E. faecalis* isolates were performed using the VITEK 2 (BioMérieux, Marcy-l'Étoile, France) system with a specific AST-GP2 card for Gram-positive bacteria. MIC results were interpreted according to the Clinical and Laboratory Standards Institute guidelines 2022.

## Genome sequencing and bioinformatic analysis

For genome sequencing, one *E. faecalis* isolate from each HBM_SA sample was selected that was purified from the MRS agar supporting probiotic bacteria growth. Briefly, genomic DNA was extracted from the fresh overnight culture of *E. faecalis* isolates using UltraClean® Microbial DNA isolation kit (Qiagen, Hilden, Germany), and genome sequencing was performed as described previously (*Yasir et al., 2020*). A Nextera DNA Flex Library Prep Kit (Illumina, Inc., San Diego, CA, USA) was utilized for library preparation, and sequencing was conducted with a V3 kit, employing 2 × 300 bp chemistry on the MiSeq platform (Illumina, Inc., USA). The quality assessment of raw sequence reads was carried out using FastQC 0.12.0, and sequence trimming was performed using the Trimmomatic v0.32 tool (*Bolger, Lohse & Usadel, 2014*). Contig assemblies were prepared using the SPAdes 3.15.3 program (*Prjibelski et al., 2020*).

Additionally, we retrieved 15 *E. faecalis* genome assemblies from the NCBI microbial genomic database (Table S1). Among them, six were isolated from HBM, seven from animal milk, and two were reference strains of Symbioflor 1 (probiotic) and V583 (vancomycin-resistant clinical strain). The *E. faecalis* genomes included in this study were from isolates recovered from the milk of healthy women and animal milk, excluding those recovered from animals with mastitis or other reported infections. Furthermore, analysis was performed from 43 *E. faecalis* isolates we previously recovered from clinical specimens in the western region of Saudi Arabia (*Farman et al., 2019*). Annotation of the genome was performed using BV-BRC 3.33.16 (*Olson et al., 2023*), and the PubMLST tool was used to determine multilocus sequence typing (MLST) (*Larsen et al., 2012*). Single nucleotide polymorphisms (SNPs) were identified in the core genomes and utilized to construct a maximum likelihood phylogenetic tree using CSI Phylogeny 1.4 with default parameters (*Kaas et al., 2014*). The phylogenetic tree was visualized using the Interactive Tree of Life (iTOL v6) tool (*Letunic & Bork, 2024*). The acquired antimicrobial resistance genes (ARGs) and chromosomal point mutations were determined employing ResFinder 4.6.0 (*Bortolaia et al., 2020*). Virulence genes were retrieved from the *E. faecalis* genomes using VirulenceFinder 2.0 (*Malberg Tetzschner et al., 2020*). Using PlasmidFinder 1.3, the origin of replications of the plasmids were identified from the assembled genomes (*Carattoli & Hasman, 2020*). Mobile genetic elements in connection to ARGs and virulence factors were determined using MobileElementFinder v1.0.3 (*Johansson et al., 2021*), and PHASTER was used to detect putative prophage elements (*Wishart et al., 2023*). Secondary metabolites and post-translationally modified peptides (RiPPs) associated gene clusters were identified in the genome assemblies using antiSMASH 7.0 (*Blin et al., 2023*). Sequence reads of isolates from this study were deposited into GenBank under BioProject ID. PRJNA1059526.

## RESULTS

### Bacterial count and community analysis in the HBM_SA samples

In the bacterial count analysis, an average of $2.3 \times 10^4 \pm 1.9 \times 10^4$ cfu/ml was obtained on sheep blood agar, and $2.7 \times 10^4 \pm 2.5 \times 10^4$ cfu/ml was obtained on R2A agar. On probiotic supporting MRS agar, $1.6 \times 10^2 \pm 75.1$ cfu/ml was obtained, whereas no growth was observed on selective MacConkey agar to detect pathogenic *E. coli*. Eighty-seven isolates were purified from the HBM_SA samples and were predominantly classified into *Staphylococcus epidermidis* and *Staphylococcus hominis*. *E. faecalis* isolates were retrieved on MRS agar in three studied samples. *Staphylococcus aureus* was found in the HMD11_46M sample, and *Staphylococcus haemolyticus* was found in the HMD9_11M sample.

### Genomic annotation of *E. faecalis* isolates

The three *E. faecalis* isolates recovered from HBM_SA samples were phenotypically resistant to erythromycin and tetracycline and sensitive to vancomycin (Table S2). Genome sequencing of three *E. faecalis* isolates, one from each HBM_SA sample, was performed. The sequenced *E. faecalis* isolates Efa_HMD9_11M, Efa_HMD11_46M, and Efa_HMD12_49M were assembled into 58, 49, and 56 contigs, respectively. The average G+C content of the isolates was $37.2 \pm 0.02\%$, and the average total length was $3,068,846 \pm 36,594$ bp (Fig. 1). According to the BV-BRC 3.33.16 (*Olson et al., 2023*) analysis, the genome quality was good, with a CheckM completeness of 100, and no CheckM contamination was detected in two of the isolates (Table S3). However, 0.5 CheckM contamination was detected in Efa_HMD9_11M. Protein-coding sequences (CDS) were detected in the range of 3,024 to 3,121. At the genomic level, the average nucleotide identity (ANI) value among the three *E. faecalis* isolates was $99.98 \pm 0.02\%$, and $99.00 \pm 0.02\%$ with the reference probiotic strains of *E. faecalis* Symbioflor 1 (Fig. 1). The ANI value was $98.8 \pm 0.3\%$ with the clinical reference strain of *E. faecalis* V583. No substantial difference was detected in the G+C content of the *E. faecalis* isolates' genomes from human milk ($37.5 \pm 0.1\%$), animal milk ($37.5 \pm 0.2\%$) retrieved from the NCBI genome database, and clinical isolates ($37.4 \pm 0.1\%$). The CDS were detected in the range of 2,580 to 3,330 (Table S3).

### *E. faecalis* genomes analysis for secondary metabolites genes

Several secondary metabolite genes were detected in the genomes of the *E. faecalis* isolates analyzed in this study and were mainly carried by the post-translationally modified peptides (RiPPs) (Table S4). Two genomic regions were found in four isolates from HBM, coding for RiPPs, including three isolates from the HBM_SA samples and the strain APC 3825 (Table S4). The lanthipeptide-class-II RiPPs were commonly found in these isolates, revealing 100% similarity with the cytolysin ClyLl/cytolysin ClyLs biosynthetic gene cluster. RiPP-like biosynthetic gene clusters (BGCs) were also detected in these four HBM isolates, showing 22% similarity with the carnobacteriocin XY (Table S4). No BGCs were detected in the genome assemblies of HBM isolates from China. Compared to isolates from human milk, cyclic-lactone-autoinducers were found in four *E. faecalis* isolates from animal milk and 20 clinical isolates. Lassopeptide (RiPPs) was also found in both animal and clinical isolates. RiPP-like (carnobacteriocin XY) was explicitly found in isolates from HBM (Table S4).

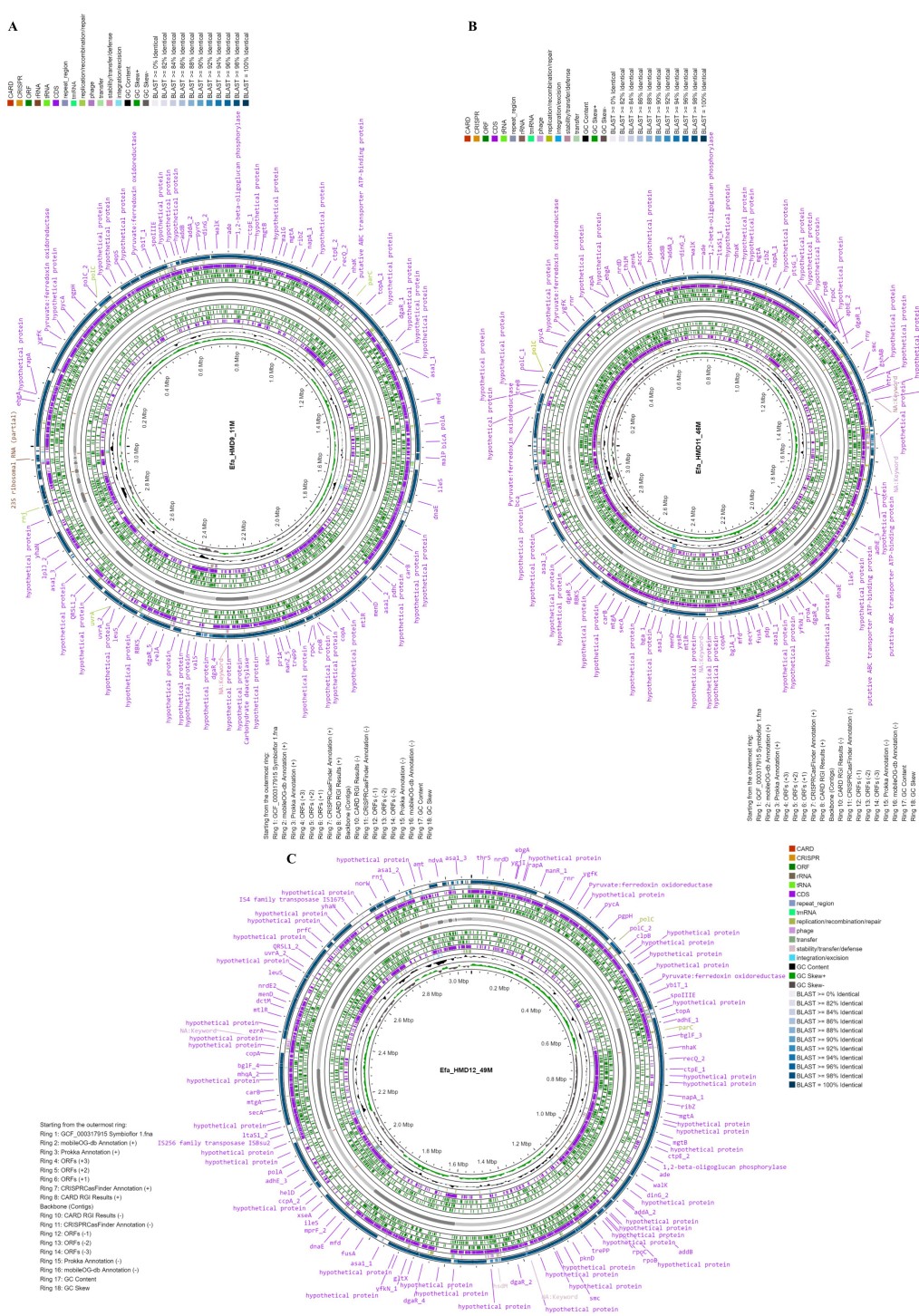

**Figure 1** **Circular maps of the three *Enterococcus faecalis* isolates with genomic features and annotation layers recovered from human breast milk samples in Saudi Arabia.** Genome map of the isolates (A) Efa_HMD9_11M, (B) Efa_HMD11_46M, and (C) Efa_HMD12_49M. The blast similarity was performed with the probiotic reference strain Symbioflor 1.

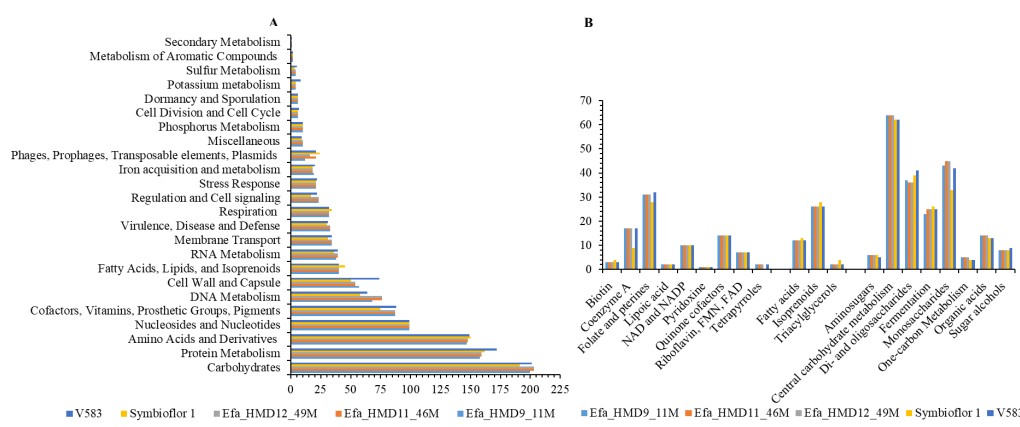

**Figure 2 Genomic annotation of the *Enterococcus faecalis* isolates recovered from human breast milk samples in Saudi Arabia and comparison with the reference strains.** (A) The dominant functional classes identified from subsystem analysis, and (B) subclass function associated with carbohydrate metabolism, fatty acid metabolism, and associated with cofactors, vitamins, prosthetic Groups. The *x*-axis in (A) and y-axis in (B) represent the number of genes associated with the relevant function. Symbioflor 1 and V583 are the reference strains.

Overall, a limited number of BGCs were detected in each genome, and no consistent pattern was found among the identified BGCs in relation to the source of isolation of these *E. faecalis* isolates.

## Comparative functional analysis of *E. faecalis* genomes

From the subsystem functions, 217 were commonly present among the genome assemblies of the three *E. faecalis* isolated from HBM_SA samples and reference strains Symbioflor 1 and V583 (Table S5). Nine subsystem functions were unique to V583, such as resistance to vancomycin and teicoplanin, potassium-transporting ATPase, toxin-antitoxin replicon stabilization, *ibrA* and *ibrB* co-activators of prophage gene expression, and aminoglycoside-modifying enzymes (Table S5). Bile hydrolysis and type I restriction-modification systems were not found in V583 but were present in Symbioflor 1 and *E. faecalis* isolates from HBM_SA samples. All detected subsystem functions were common in the HBM_SA *E. faecalis* isolates (Table S5). Moreover, inositol catabolism, formaldehyde assimilation (ribulose monophosphate pathway), and tetracycline resistance were unique to isolates from HBM_SA samples and were not detected in both reference strains (Table S5).

In comparing HBM_SA isolates with other HBM isolates from China, 224 out of 230 subsystem functions were commonly present, whereas 229 subsystem functions found in the APC 3825 isolate from Ireland were commonly found in the HBM_SA isolates (Table S5). Interestingly, the five *E. faecalis* isolates from HBM in China commonly shared all the 226 detected subsystem functions. Subsystem functions like hyaluronate utilization, inositol catabolism, formaldehyde assimilation, and tetracycline resistance were uniquely detected in the HBM_SA and APC 3825 isolates compared to HBM isolates from China. The hyaluronic acid capsule and hyaluronic acid-containing cell wall-associated genes were found in HBM isolates from China and not in the HBM_SA isolates. In total, 190 out of

248 subsystem functions were common in the isolates recovered from human and animal milk, whereas 214 out of 259 subsystem functions were common between HBM_SA isolates and clinical isolates recovered from patient specimens in Saudi Arabia (Table S5). Overall, 188 out of 263 subsystem functions were common in all the analyzed genome assemblies (Table S5).

In comparing gene families from the CAZy database, which catalogs microbial enzymes catalyzing carbohydrates, we identified 43 CAZy families and sub-families that were common in the HBM_SA isolates. Among these, 35 were shared with both reference strains, Symbioflor 1 and V583 (Table S6). The GH24 family was specifically found in the HBM_SA isolates. In contrast, seven CAZy families and sub-families (CE4, GH136, GH154, GH31, GH88, GT26, PL12_1) were common among the HBM_SA isolates and V583 strains but were not detected in the Symbioflor 1 genome (Table S6). As a result of the functional annotation conducted with RAST, carbohydrate metabolism was the most enriched metabolic category in the *E. faecalis* isolates from the HBM_SA samples and in both reference strains (Fig. 2A). These isolates exhibited the ability to metabolize di- and oligosaccharides, monosaccharides, and organic acids (Fig. 2B). The second-highest percentage of PEGs was in the protein metabolism category, followed by amino acids and derivatives, and nucleosides and nucleotides (Fig. 2A). Genes encoding for lipoic acid, pyridoxine, riboflavin, FMN, FAD, and tetrapyrroles from the subclass cofactors, vitamins, prosthetic groups, pigments, as well as genes for fatty acids, isoprenoids, and triacylglycerols, were found in the HBM_SA isolates and both reference strains (Fig. 2B).

## Phylogenetic and multilocus sequence typing analysis

In a phylogenetic tree based on SNPs, HBM_SA isolates were clustered together, and several clinical isolates previously recovered from patient specimens in Saudi Arabia (Fig. 3). The isolates from HBM in China were clustered along the QH5 isolate from Yalk milk with Symbioflor 1. Overall, the isolates from HBM and animal milk clustered distinctly from the highly characterized virulent strain V583 (Fig. 3). The MLST analysis classified three *E. faecalis* isolates from HBM_SA samples and the APC 3825 strain isolated from HBM in Ireland as sequence type ST179 (Fig. 4A). Five *E. faecalis* isolates from HBM in China were classified as ST25. Variations were found in the sequence types of *E. faecalis* isolates from animal milk and clinical samples. None of the isolates from animal milk were classified as ST179 or ST25, whereas 11 isolates from clinical specimens were classified as ST179 (Fig. 4A). The probiotic reference strain Symbioflor 1 was classified as ST248, and the clinical reference strain for vancomycin resistance, V583, was classified as ST6 (Fig. 4A).

## Comparative antimicrobial resistance genes analysis

In isolates from HBM, the fewest number of ARGs were identified. The *lsaA* gene, intrinsic to *E. faecalis* and conferring resistance to lincomycin, clindamycin, dalfopristin, pristinamycin IIA, and virginiamycin M, was commonly found in the studied isolates (Fig. 4A). The *tetM* and *ermB* genes, responsible for tetracycline and macrolide antibiotic resistance, were detected in the HBM_SA isolates. *E. faecalis* HBM isolates from China carried only the *lsaA* gene, similar to the Symbioflor 1 strain (Fig. 4A). The APC 3825 isolate carried *tetM*

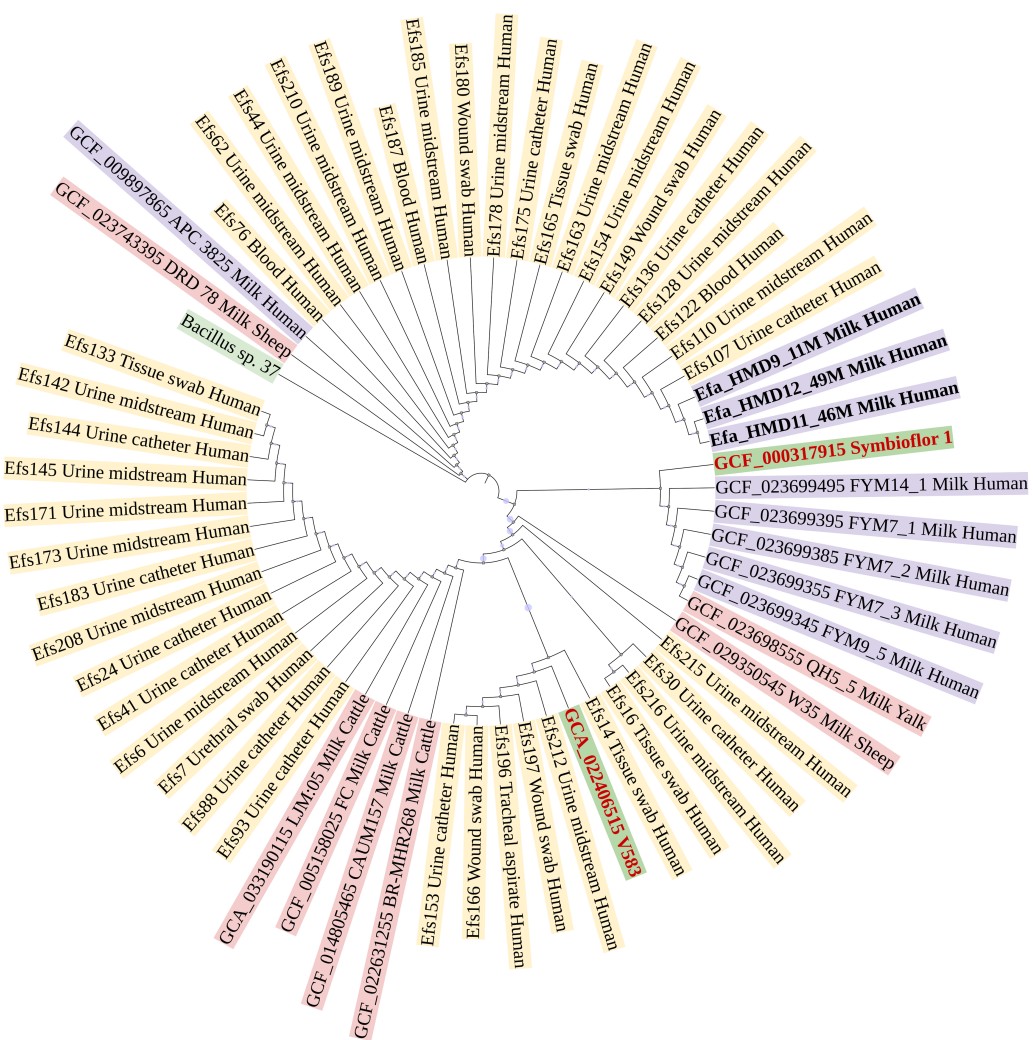

**Figure 3** **Phylogenetic linkage among *Enterococcus faecalis* isolates based on SNPs in core genomes.** The sequenced isolates from this study are mentioned in bold font. Along with strain identification, the host and source of isolation are mentioned in the tree. Symbioflor 1 is a probiotic, and V583 is a clinical reference strain.

gene. However, the ARG pattern of *E. faecalis* isolates from animal milk of sheep and yalk resembled that of HBM isolates (Fig. 4A). Two isolates from cattle milk carried multiple resistance genes. In the isolates from clinical samples, 30 isolates were carrying ≥6 ARGs, conferring resistance to clinically important antibiotics from the classes of aminoglycoside, phenicol, diaminopyrimidine, and macrolide. Furthermore, no mutation was detected in the *gyrA* and *parC* genes, causing quinolone resistance in the HBM_SA isolates, similar to the Symbioflor 1 strain (Fig. 4B). However, the known parC p.S517N mutation was detected in HBM isolates from China (Fig. 4B). Most isolates from animal milk did not carry mutations in the *gyrA* and *parC* genes (Fig. 4B). The LJM05 strain from cattle carried an unknown mutation parC p.V13I (GTA − > ATA, V − > I). In clinical isolates, parC

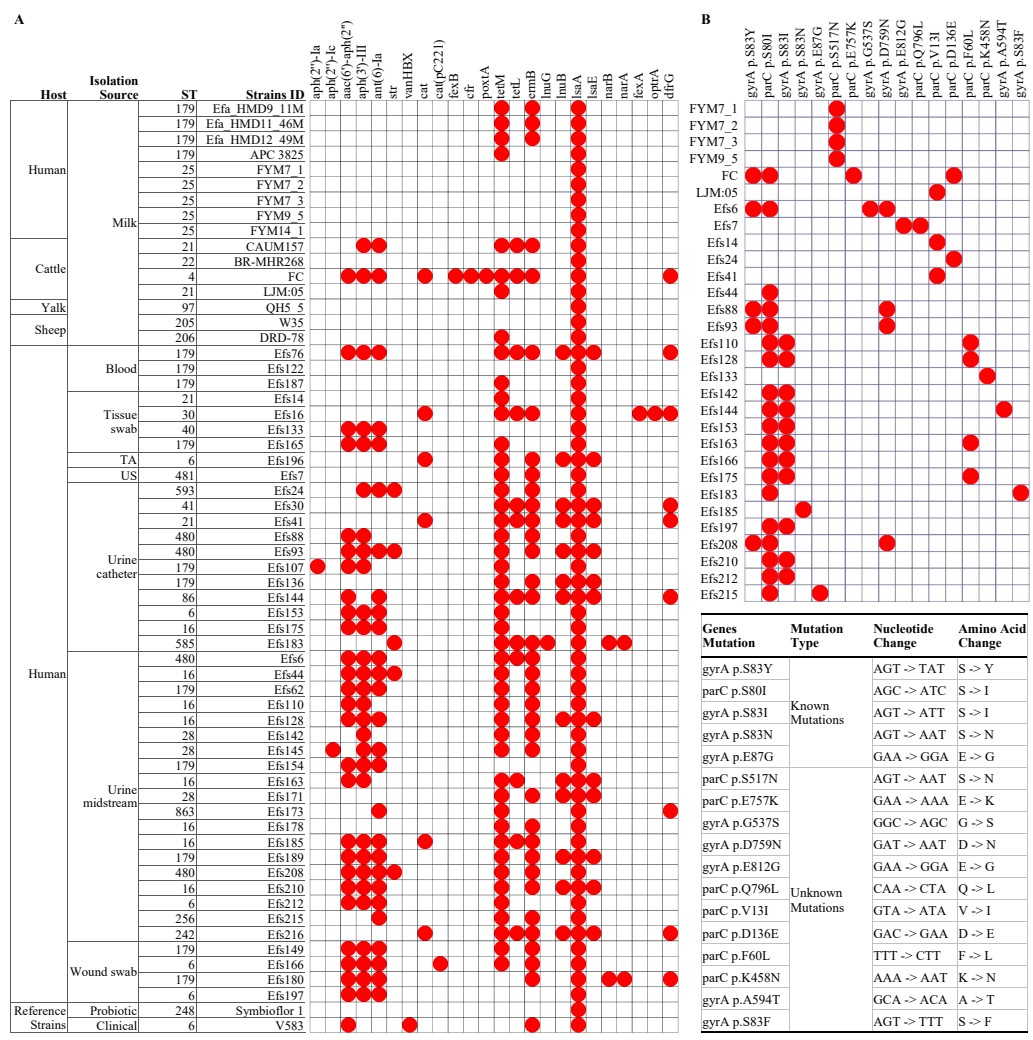

**Figure 4** **Multilocus sequence typing and identification of antimicrobial resistance genes from genome assemblies of *Enterococcus faecalis* isolates.** (A) Analysis of antimicrobial resistance genes, and (B) identification of known and unknown mutations in the gyrA and parC genes. The white boxes indicate the respective gene or mutation was not detected in (A) and (B). TA, tracheal aspirate; US, urethral swab.

p.S80I mutation was detected in 18 isolates, and gyrA p.S83I mutation was detected in 11 isolates (Fig. 4B). Additionally, ten unknown mutations were detected in several clinical isolates (Fig. 4B).

## Virulence factors associated genes analysis

In comparing virulence genes in *E. faecalis* isolates from HBM, 13 genes were commonly found in isolates from HBM_SA samples, APC 3825, and HBM isolates from China (Fig. 5). However, the *cylM*, *cylL*, *cylA*, and *acm* genes were detected in the HBM_SA isolates and APC 3825 but not in the HBM isolates from China (Fig. 5, Table S1). Compared to the reference probiotic strain, 12 virulence genes detected in HBM isolates were common to

Symbioflor 1. The virulence gene profiles in the isolates from animal milk matched those of HBM isolates, with ten virulence genes commonly detected among the isolates from HBM and animal milk (Fig. 5). In clinical isolates, more variation was noticed in the virulence genes detected in the 13 to 22 genes range (Fig. 5). The virulence genes *fsrB* and *hylB* were found in clinical reference strains V583 but were not detected in the HBM_SA isolates. An operon of three genes, *ebpA*, *ebpB*, and *ebpC*, encoding the pilus subunits of PilB, was found in the genome of *E. faecalis* clinical isolates, but in the HBM isolates, *ebpB* was not detected. Interestingly, genes encoding exoenzymes and toxins like gelatinase (*gelE*) were commonly found in most isolates, but the hyaluronidase gene *hylB* was not found in the HBM isolates and was detected in 16 clinical isolates and V583 (Fig. 5). The gene *espfm*, involved in the virulence and biofilm-forming capacity, and *fsrB*, involved in the quorum-sensing system of *E. faecalis*, were not detected in the HBM isolates (Fig. 5).

## Prophages, and mobile genetic elements analysis

In the analysis of putative prophage elements using PHASTER, intact phiFL1A was commonly found in HBM_SA *E. faecalis* isolates but was not present in intact form in either of the reference strains. The intact phage EFC_1 and phiFL4A were found in Efa_HMD11_46M. The phiFL3A and phiFL4A were detected in Efa_HMD12_49M. The V538 strain was carrying phiFL4A and phBC6A52. The probiotic Symbioflor 1 genome carried four intact phages of PBL1c, phiFL2A, phiEf11, and phBC6A52. The mobile genetic elements of insertion sequence ISLgar5, ISSsu5, Tn6009, and composite transposon cn_11752_ISLgar5 were commonly found in the three HBM_SA isolates, but those were not found in the Symbioflor 1 strain, and ISLgar5 was detected in the V583 strains along with five other MGEs (Table S7).

## Plasmids and associated antibiotic resistance genes analysis

A diversity of plasmids was found in the analyzed 61 genomes of *E. faecalis* isolates, classified into 15 different replicons (Fig. 6). The HBM_SA isolates commonly carried the replicons repUS43 and rep9b. The rep9a replicon was not detected in the Efa_HMD9_11M isolate. No plasmids were detected in the HBM isolates from China and Symbioflor 1 (Fig. 6). The repUS43 replicon was the most common replicon type identified, also prevalent in animal milk (four out of eight, 50%) and clinical isolates (36 out of 43, 83.7%), followed by rep9b detected in 17 out of 43 clinical isolates (Fig. 6). The *tetM* gene from HBM_SA isolates were found on the repUS43 replicon carrying plasmid, which carried transposon Tn6009. The *ermB* gene was found on a contig carrying the IS1380 insertion sequence in the Efa_HMD11_46M and Efa_HMD12_49M isolates.

## DISCUSSION

HBM microbes shape the infant's gut microbiota and impact the overall health of the child (*Le Doare et al., 2018*). The microbiota of HBM facilitates beneficial commensal flora growth to enhance the infant's immune system and reduce enteropathogens colonization (*Le Doare et al., 2018*). In this study, we adopted a culturing method to assess HBM aerobic bacteria in the Saudi population and evaluated the probiotic potential and

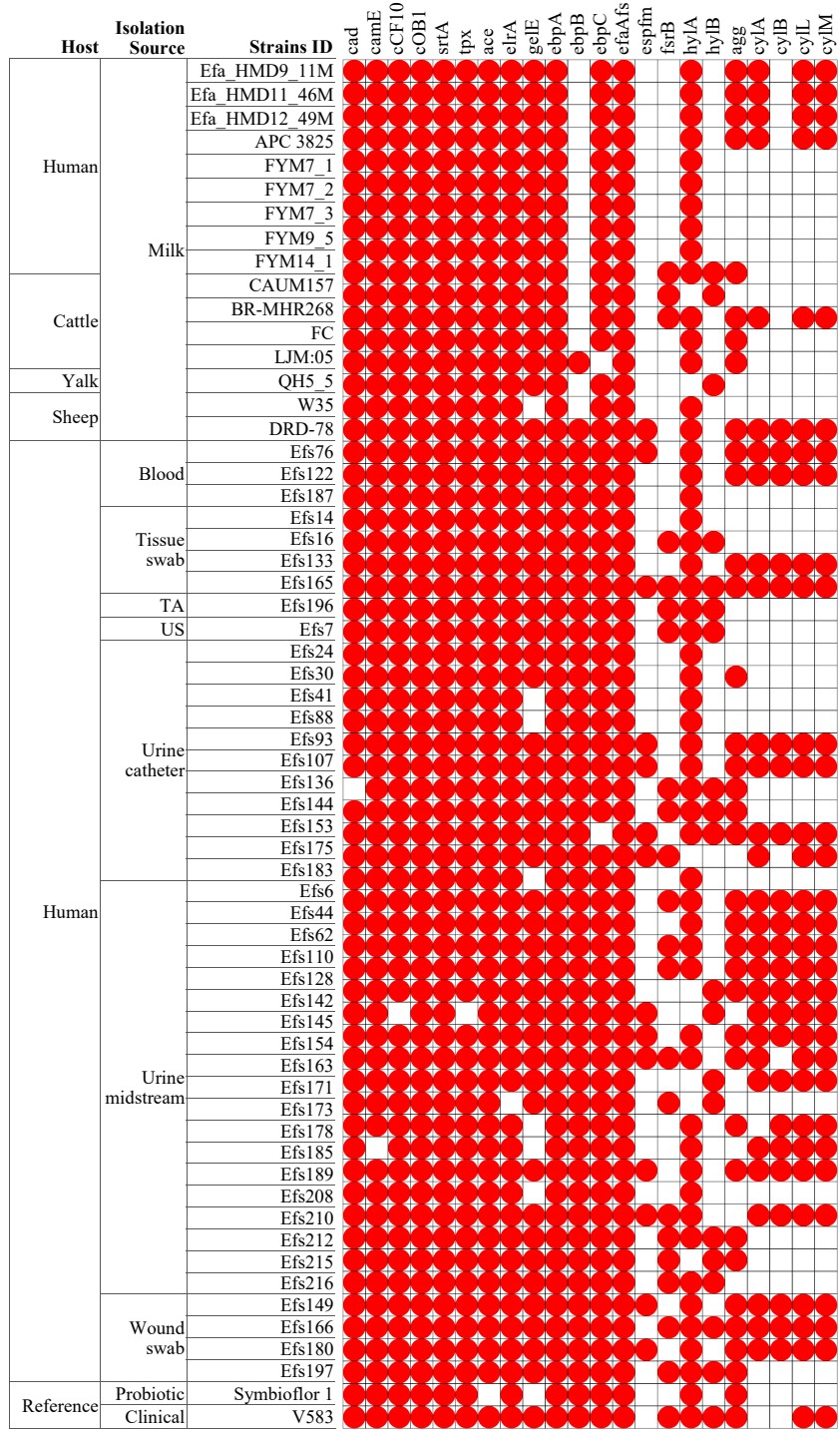

**Figure 5 Distribution of virulence factors associated genes identified in the *Enterococcus faecalis* genome assemblies.** The white box indicates the respective genes were not detected. TA, tracheal aspirate; US, urethral swab.

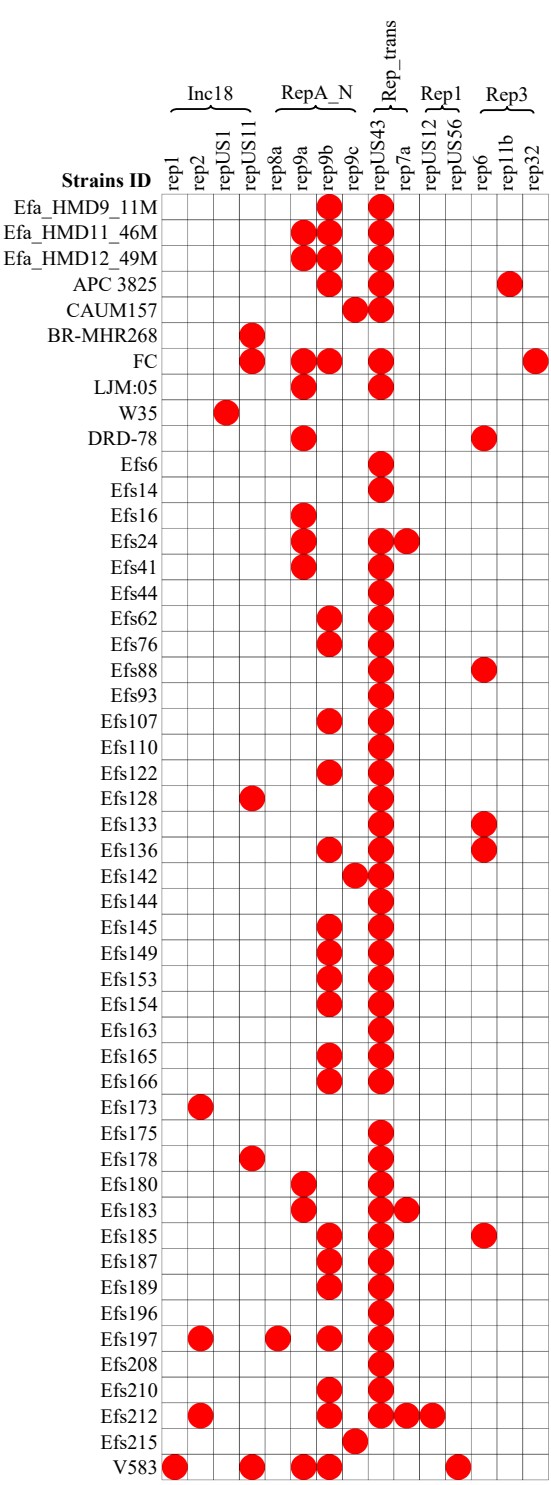

**Figure 6 Distribution of plasmid replicon found in the *Enterococcus faecalis* genome assemblies.** The white box indicates the respective replicons were not detected.

safety of *E. faecalis* isolates from genomic analysis. The low number of HBM samples analyzed in this study is a limitation of this study. However, *E. faecalis* genomes were retrieved from databases for comprehensive analysis. The results revealed the presence of *S. aureus*, *S. epidermidis*, *E. faecalis*, and *S. hominis* in HBM_SA samples. The detection of staphylococci, enterococci, and streptococci in HBM is quite common, and *E. faecium* and *E. faecalis* presence in HBM samples of healthy women has been described in various studies (*Anjum et al., 2022*; *Bagci et al., 2019*; *Jost et al., 2013*; *Reis et al., 2016*). Similarly, enterococci species (*E. durans*, *E. faecalis*, *E. casseliflavus*, *E. faecium*, and *E. hirae*) have been detected in healthy cattle and sheep milk samples (*McAuley et al., 2015*; *Souza et al., 2023*). *Albesharat et al. (2011)* identified LAB strains (*L. fermentum*, *L. plantarum*, *E. faecium*, and *E. faecalis*) with identical genotypes in the breast milk and feces samples of infants and mothers, suggesting the transmission of these species from mother to infant.

This study also evaluated total bacterial counts (TBCs) in HBM samples of healthy donors. A varying range of TBC counts ($8.0 \times 10^2$ to $3.9 \times 10^4$ CFU/ml) was observed in HBM_SA milk samples. Notably, *S. aureus was* only detected in one HBM sample (HMD11_46M) of an asymptomatic healthy donor. Unlike raw milk regulations of dairy farms, TBC levels have not been specified for human milk to restrict the breastfeeding of infants (*Jayarao et al., 2004*). Generally, a TBC count of less than $10^3$ cfu/ml is acceptable regardless of the type of organisms, whereas a TBC count of more than $10^5$ cfu/ml is considered unsafe (*Balmer & Wharton, 1992*). Donor milk TBC counts within a range of $10^3$ and $10^5$ CFU/ml are accepted if only skin commensal microbes (viridans streptococci, diphtheroids, and *S. epidermidis*) are present (*Balmer & Wharton, 1992*). There are no TBC restrictions for human breastfeeding, and only human T-cell leukemia virus type I, HIV, and cytomegalovirus-infected mothers are suggested to stop breastfeeding their infants (*Huang et al., 2019*; *Lawrence & Lawrence, 2001*; *Li et al., 2004*). Another perspective is that a high TBC count might also contribute to the presence of diverse bacterial strains in a healthy infant's gut (*Fernandez et al., 2013*).

*E. faecalis* and *E. faecium* are recurrent in HBM, but some studies have also reported the presence of only one of these species in human milk (*Huang et al., 2019*; *Jimenez et al., 2008*; *Khalkhali & Mojgani, 2017*; *Kivanc, Kivanc & Yigit, 2016*). During this study, *E. faecalis* isolates were detected in HBM_SA samples but *E. faecium* isolates were not retrieved. The glycome content of some mother's milk allows the flourishing of enterococcal spp. whereas lower enterococci numbers are correlated with poor infant health (*Korpela et al., 2018*; *Laursen et al., 2020*). Human milk microbiota might originate from the maternal intestine where they are present in large numbers (*Jimenez et al., 2008*; *Jost et al., 2013*; *Rodriguez, 2014*). However, several HBM *E. faecalis* strains are also considered unsafe, which otherwise are promising probiotic bacteria (*Khalkhali & Mojgani, 2017*). This study conducted a comparative genomic analysis of the pathogenic and probiotic potential of HBM-isolated *E. faecalis* strains (*Anjum et al., 2022*; *Huang et al., 2019*; *Khalkhali & Mojgani, 2017*). The analyzed *E. faecalis* isolates did not differentiate regarding the average number of subsystem functions associated genes, GC content, and genome size. Their genome size ranged between 2.7 and 3.2 Mb, whereas GC contents were noted to be 37.1% to 37.7% (Figs. 1A–1C, Table S3). Consistent to the findings of this study, the genomic

similarities have been reported in *E. faecalis* strains irrespective of the host, temporal, geographical, and non-clinical/clinical origins (*Bonacina et al., 2017*). Gene gain/loss events or genomic fluidity in *E. faecalis* has been linked to *E. faecalis* ability to acquire external drug-resistance genes (*Panthee et al., 2021*).

The probiotics should lack transferable antibiotic resistance and pathogenic genes to confer health benefits (*Fernandez et al., 2015*). These features are strain specific. Therefore, each strain should be subjected to individual genetic evaluation for probiotic applications (*Krawczyk et al., 2021*). Unlike established probiotic bifidobacteria and lactobacilli, the enterococci lack global safety certifications including QPS (qualified presumption of safety) status from EFSA (European Food Safety Authority) and GRAS (Generally Recognized as Safe) status from US-FDA (US Food and Drug Administration) (*Hanchi et al., 2018*). Data from this study revealed the presence of 11 virulence genes in more than 95% ($n = 59/61$) of analyzed genomes irrespective of isolation source. Previously, a high prevalence of virulence factors such as adhesion factors-encoding genes (*esp, efaA,* and *agg*) was reported in *E. faecalis* isolates (*Dapkevicius et al., 2021*; *Krawczyk et al., 2021*). The *gelE* and *efaAfs* genes were present in the HBM isolates and most of the other isolates from animal and clinical samples (Fig. 5). In addition to the *gelE* genes, other factors are also required for gelatinase functioning (*De Castilho, Nero & Todorov, 2019*). The *fsr* transcription regulator gene's product activates *gelE* expression, which is crucial for enterococcal bacteria functioning (*Dundar et al., 2015*). The results of this study revealed the absence of the *fsrB* gene in HBM isolates. The *ace* gene was commonly detected in HBM and clinical isolates except *Symbioflor 1,* which contributes to *E. faecalis* colonization in the heart valves and may cause endocarditis (*Silva, Montalvao & Bonafe, 2017*).The ebp operon-encoded pili (endocarditis-associated pili) also contribute to biofilm formation and binding of thrombocytes, collagen, and fibrinogen (*Krawczyk et al., 2020*). The results showed the presence of pilus subunits of the PilB-encoding operon of three genes (*ebpC, ebpB*, and *ebpA*) in clinical *E. faecalis* genome but *ebpB* was not detected in HBM isolates, which suggested its inactivation. The absence of regulatory genes for virulence factors in *E. faecalis* isolates from the HBM sample reduces their safety concerns as a probiotic. Moreover, some virulence factors, apart from their role in pathogenicity, are also significant for probiotic potential, particularly those associated with adhesion and colonization. For example, capsule-forming and adhesion-associated genes were identified in all HBM *E. faecalis* isolates, which are important for the adhesion, colonization, and capsule formation of commensal probiotic enterococci inside the host (*Pillar & Gilmore, 2004*). Moreover, capsule formation and adhesion–related genes have also been reported in probiotic/starter strains of enterococci (*Baccouri et al., 2019*).

Cytolysin is a two-subunit toxin that activates in response to the sensation of the host cell. Several *E. faecalis* virulent strains produce it to damage eukaryotic tissues and thus contribute to pathogenesis (*Coburn et al., 2004*). *cylA, cylM*, and *cylL* genes were detected in HBM_SA isolates during this study. Contrarily, these genes were not found in Chinese HBM isolates and Symbioflor 1 strains. However, the *cylB* gene was not detected in HBM isolates, whereas it was found in the genomes of several clinical isolates. Hyaluronidase presence enhances the clinical strain's virulence. It affects hyaluronic acid, leading to

the breakdown of connective tissue *via* mucopolysaccharide moieties' depolymerization (*Krawczyk et al., 2021*). Hyaluronidase encoding gene (*hyl*) reported in several pathogenic enterococci. During the current study, *hylA* was commonly detected in HBM and other clinical isolates, whereas the Hyaluronate lyase-encoding *hylB* gene remained absent. The presence of *hyl, cyl,* and *gel* genes has often been reported in lower frequency in foodborne strains than clinical enterococci (*İspirli, Demirbaş & Dertli, 2017*; *Krawczyk et al., 2021*). Overall, certain virulence-impacting factors of enterococci (proteolytic system, aggregation factors, and exopolysaccharide (EPS) production) can be advantageous in probiotic strains (*Krawczyk et al., 2021*; *Ramos & Morales, 2019*). For instance, EPS synthesis facilitates the movement of non-motile bacterium toward nutrient-rich environments and helps in escaping stressful conditions (osmolarity, higher pH, and temperature), toxicity (pancreatic enzymes, antibiotics, gastric, metal ions, and bile salts), and human immune responses (*Lynch et al., 2018*). Similarly, virulence genes are previously reported in the genomes of generally recognized as safe (GRAS) probiotics like *Lactiplantibacillus plantarum* and *Lactiplantibacillus pentosus*, which play a role in helping pathogenic bacteria adapt, survive or adhere in hostile or host environments (*Chokesajjawatee et al., 2020*; *Stergiou et al., 2021*). However, in the absence of other pathogenic mechanisms, these genes can be considered advantageous for the bacterium, as they enhance bacterial fitness and may be beneficial in contexts where live cells are required (*Chokesajjawatee et al., 2020*).

The reports of multiple antibiotic resistance in enterococci have increased recently (*Farman et al., 2019*; *Huang et al., 2019*; *Leigh et al., 2022*). They can acquire and share adaptive traits such as mobile genetic elements-encoded antimicrobial resistance-related genes (*Leigh et al., 2022*; *Mikalsen et al., 2015*). Therefore antibiotic resistance and susceptibility of potential probiotic strains of *Enterococcus* should be thoroughly assessed. During this study, HBM_SA isolates presented a lower number of acquired resistance genes in addition to intrinsic gene *lsa(A)* (Fig. 4), which is in agreement with previous studies (*Jimenez et al., 2008*; *Krawczyk et al., 2021*). Antibiotic overuse (tetracycline and macrolide) against enterococcal infection may lead to acquired enterococci resistance *via tetM* and *ermB* genes' horizontal transfer (*De Leener et al., 2004*). In a previous study, HBM-isolated *Enterococcus* spp. from healthy mothers demonstrated resistance against different clinical antibiotics (*Huang et al., 2019*). Aligned with our findings, specific antibiotic resistance determinants carried on mobile genetic elements, such as tetracycline resistance genes, are found in probiotic genera, acting as a reservoir of resistance for potential foodborne or gut pathogens and posing a safety risk (*Gueimonde et al., 2013*). Overall, the antibiotic susceptibility of HBM isolates was more related to the probiotic strain Symbioflor 1 than the aminoglycoside resistance gene-carrying classical *E. faecalis* clinical strains.

The genome annotation of HBM_SA-isolated *E. faecalis* and other *E. faecalis* isolates revealed that CDS assigned to subsystems for carbohydrate metabolism was the major metabolic category. These findings are in line with the growth capability of enterococci under different environments by metabolizing various types of sugar (*Ramsey, Hartke & Huycke, 2014*). HBM isolates did not exhibit certain exclusive functions and their roles were similar to clinical specimens. Notably, 188 of 263 analyzed subsystem functions were present in all the analyzed isolates and thus can be attributed to the basic species function.

The utilization of ethanolamine by *E. faecalis* was commonly noted in analyzed genomes that help in the survival of gastrointestinal tract-inhabiting bacteria and contribute to pathogen virulence (*Fox et al., 2009*).

Antimicrobial capability is a highly desirable feature of probiotics, which empowers the bacteria to eliminate pathogenic infections (*Bagci et al., 2019*; *Reis et al., 2016*). Enterococci can protect the neonatal gut from various pathogens (high-risk enterobacteria, *Salmonella* spp., *S. aureus*, and *S. pyogenes*) to ensure HBM safety (*Al Atya et al., 2015*; *Fernandez et al., 2020*). Similarly, the broad-spectrum anti-pathogenic activity of newborn gut-isolated *E. faecalis* strains, which was activated through bacteriocin-like inhibitory substances, has been reported (*Al Atya et al., 2015*). The study also explored secondary metabolites biosynthesis-associated gene clusters in *E. faecalis* genomes. A putative SM gene cluster was identified in HBM_SA isolates, which harbored ribosomally synthesized and post-translationally modified peptide products (RiPPs) that have been previously reported in *E. faecalis* isolates (*Panthee et al., 2021*). Several SM gene clusters were identified in the *E. faecalis* genome. Most were predicted to be either peptides, peptide-related compounds, or RiPP-like compounds, which enhanced the antimicrobial and probiotic capability of *E. faecalis* isolates.

## CONCLUSIONS

The virulence factors of HBM and animal milk-isolated *E. faecalis* indicate a safety concern, despite their potential probiotic features. The antimicrobial resistance of these isolates was comparatively lower than the vancomycin-resistant strain (*E. faecalis* V583) and other clinical isolates. The HBM isolates from China were more closely aligned with the probiotic strain Symbioflor 1. The study recommends individual safety assessment of each *E. faecalis* isolate for probiotic validation, utilizing both phenotypic analysis and genomic characterization. We suggest conducting further studies with larger sample sizes.

### Funding

This project was funded by the Deanship of Scientific Research (DSR) at King Abdulaziz University, Jeddah, under grant no. (GPIP: 1623-141-2024). The funders had no role in study design, data collection and analysis, decision to publish, or preparation of the manuscript.

### Grant Disclosures

The following grant information was disclosed by the authors:
Deanship of Scientific Research (DSR) at King Abdulaziz University, Jeddah: GPIP: 1623-141-2024.

### Competing Interests

The authors declare there are no competing interests.

## Author Contributions

- Lobna Badr performed the experiments, analyzed the data, prepared figures and/or tables, and approved the final draft.
- Muhammad Yasir conceived and designed the experiments, performed the experiments, analyzed the data, prepared figures and/or tables, authored or reviewed drafts of the article, and approved the final draft.
- Areej A. Alkhaldy conceived and designed the experiments, performed the experiments, authored or reviewed drafts of the article, and approved the final draft.
- Samah A. Soliman performed the experiments, prepared figures and/or tables, and approved the final draft.
- Magdah Ganash conceived and designed the experiments, authored or reviewed drafts of the article, and approved the final draft.
- Safaa A. Turkistani performed the experiments, analyzed the data, authored or reviewed drafts of the article, and approved the final draft.
- Asif A. Jiman-Fatani conceived and designed the experiments, authored or reviewed drafts of the article, and approved the final draft.
- Ibrahim A. Al-Zahrani analyzed the data, authored or reviewed drafts of the article, and approved the final draft.
- Esam I. Azhar conceived and designed the experiments, authored or reviewed drafts of the article, and approved the final draft.

## Human Ethics

The following information was supplied relating to ethical approvals (*i.e.*, approving body and any reference numbers):

Dr. Soliman Fakeeh Hospital, Jeddah, Saudi Arabia.

## Data Availability

The sequence reads of isolates from this study are available in GenBank: PRJNA1059526.

## Supplemental Information

Supplemental information for this article can be found online at http://dx.doi.org/10.7717/peerj.18392#supplemental-information.

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
