# Peer review of "Genomic evaluation of the probiotic and pathogenic features of Enterococcus faecalis from human breast milk and comparison with the isolates from animal milk and clinical specimens"

_PeerJ, doi:10.7717/peerj.18392_

## Round 0.1 · original submission · Minor Revisions

Reviewers found your article interesting and relevant, but the majority had concerns about certain sections of the manuscript. These will need addressing before the manuscript can be accepted.

·

Basic reporting

No comment (see attachment)

Experimental design

The experimental design was well executed. See document attached

Validity of the findings

No comment (see attachment)

Additional comments

No comment (see attachment)

·

Basic reporting

.

Experimental design

.

Validity of the findings

.

Additional comments

I reviewed the article entitled "Genomic evaluation of the probiotic and pathogenic features of Enterococcus faecalis from human breast milk and comparison with the isolates from animal milk and clinical specimens".
In my idea this article has valuable data and is well written.
It can be published in the journal without any changes.

Reviewer 3 ·

Basic reporting

the language use was Clear there were some ambiguity and some minor language issues but overall the manuscript was clear intention and description of the work.

The following language ambiguities were noted.

• Line one of the abstract: feels like it contradicts itself and could be better phrased to reflect that E.f. can be both considered a potential probiotic, commensal but there are circulating AMR and virulence determinants that could pose a risk in some stains.
• Abstract mentions clinical specifying it could help to indicate these are wound and urinary clinical specimens that would improve accessibility to wider audience.
• Line end of line 78 into 79 may read better The [combination of] virulence… and this sentence could be reviewed with this in mind for clarity.
• Line 111 [sterile] conditions… not sterilization.
• Line 141 would be better if they describe sources of these in text to give context to reader for accessibility software for reading.
• Line 150 ……plasmids [were] identified…. as plasmids is plural not was a plural
• Line 185 Several secondary metabolites? Should this be Several secondary metabolic genes…. Similarly the same point for the section titles line 184.
• Line 193 Compared to [isolates from] human milk
• Line 193 it would also help if clear this is for both HBM isolates form this study and previous study?


The literature review and referencing throughout was suitable for the work presented.

The structure and figure presentations were clear and of suitable nature.

I was unclear if figure 1 added to the manuscript if an annotated sequence is uploaded to sequence repositories.

Experimental design

The authors work is in scope and presents data on isolation of three human breast milk isolated Enterococci. They sequenced strains. Their aims was clear to compare these sequences to those previously described some from human breast milk from a different study and other from clinical isolates mainly wood and urinary and one ‘virulent’ isolate.

The analysis is well done and reviews metabolic genes, AMR, presence of virulence and compare the strains by phylogenetic analysis. They have considered MLST but I felt it was unclear how this was linked to the wider phylogenetic comparisons.

As such this included some limited new data but also some meta-analysis of existing data for comparison.



The ethical review and integrity of the work and good. There are minor language issues these could be improved

Validity of the findings

The data analysis is sound but I did feel there was some weakness in sample numbers form their isolates making some of the discussion statements weaker as there could be potential for bias in interpretation due to the small number of samples. This is observable by the difference between other previously published HBM strains compare to the one isolated as part of this study which segregated differently in the phylogenetic and other difference as note din the report.

Therefore, the analysis and interpretation could be stronger.

For me the discussion ws almost too general this need to be more nuanced and specific on what the data meant like it reviewed the data but was not critical of weakness of limited source data of only three HBM samples from their study. I
this means the conclusion ”…makes them an intermediate species” needs a more clear cut argument and or more samples and data.

So for me it felt quite weak as it was only based on limited data. This needs to be comment on or more data included.

Additional comments

The data analysis was appropriate and cover a lot of ground and I felt the overall approach was useful a descriptive approach. The understanding of how different sourced enterococci interrelate and how you risk assess them as potential probiotics is useful.

Reviewer 4 ·

Basic reporting

Introduction - no comments.

Methods – The methods section requires more attention to detail to ensure reproducibility. Some tools have not been cited and versions for some are missing. I suggest that that these details are added.

Tables – I would suggest adding a new column defining if the samples were isolated in this study or downloaded from other databases. This might help with the confusion over which samples are being assessed in the results section. A legend for most tables is missing, please add this to help with comprehension.

Discussion – The discussion is mostly background and detracts from the importance of the results. Figure and table numbers would help support the discussion and make stronger arguments.

Experimental design

The study has been designed well. However, specific references and versions of tools used is frequently missing. This impacts on the reproducibility of the study.

There is little mention to exclusion and inclusion criteria for isolates/samples used in this study.

Validity of the findings

The study primarily focuses on the results of HBM samples and often overlooks the results from the downloaded dataset. This links to the comments about inclusion criteria.

Additional comments

General comments:

Badr et al. have undertaken a comprehensive analysis of clinical, commensal and probiotic Enterococcus faecalis isolates from human and animal origin. The authors argue that evaluation of the pathogenic and resistance features of this bacterium must be considered before designating strains for use as probiotics. This is important due to the pathogenic potential of this bacteria.

Methods – The methods section requires more attention to detail to ensure reproducibility. Some tools have not been cited and versions for some are missing. I suggest that that these details are added.

Tables – I would suggest adding a new column defining if the samples were isolated in this study or downloaded from other databases. This might help with the confusion over which samples are being assessed in the results section. A legend for most tables is missing, please add this to help with comprehension.

Discussion – The discussion is mostly background and does not discuss the importance of the results clearly. Figure and table numbers would help support the discussion and make stronger arguments.

Comments on specific lines:

71-73. Please correct in-text citation for this sentence e.g. “Anjum et al. (2022) have recently. “

79 – what is meant by “simple.”

110 – an average is one single number. Is the +- 2.6 years the range?

115-116 – it is not entirely clear what medium was used for the serial dilutions. Was this broth of some description, PBS, etc.? Without this information the methodology will not be reproducible.

117-118 – More information is required to ensure this methodology is reproducible. What is the rest of the cryo storage medium made from.

121-122 – please explain how the Escherichia coli strain was used to validate the VITEK-MS run when detecting E. faecalis.

122-125 – Were these ASTs validated using a control strain?

134 – what version of FastQC was used. Please also site this tool.

135 – please site the Trimmomatic tool.

137-138 – It is not clear in table S1 which 15 genomes you refer to in this passage of text. Table S1 contains 61 different strain accessions. Please make this clear to the reader which strains you are referring to.

138-141- related to the previous comment, it is not clear which strains these relate to in table S1. Furthermore, there are 61 strains in the table but only 57 isolates are described in lines 137-156. Please adjust the text or table to make it clearer to the reader for which strains you are referring to in this section.

144-145. Please explain how SNPs were identified, what thresholds were used. This is required for reproducibility.

146 – please cite the iTOL tool.

148 – please cite ResFinder.

152 – what versions were used for these tools.

156 – PRJNA1059526 is not yet active on the NCBI, please ensure the data is published.

159 – Similar to line 110. The average should be one number. Is the +- 1.9 X104 the range? I suggest adding the range in brackets and stating what this refers to.

160 – please see lines 110 and 159 comments. Please correct throughout the manuscript (such as lines 161, etc.)

163 – please explain what is meant by “mainly.”

164-165 – which sample numbers do these refer to.

174-175 – please cite this tool and define the version used.

117 – It is unclear how a range of proteins is detected from one sample. Should this number not be fixed? How does this number relate to the expected number of CDS in the target bacteria.

177 – what does ANI stand for?

175-177 – The contamination score is higher for other strains in table S3. Why has only stain Efa_HMD9_11M being highlighted by this metric.

Table S3 – the checkM completeness is missing for sample V583.

176-177 – Again, the range is only denoted here for 3 isolates and does not consider the other ~ 60 isolates in this study. Please explain why the other strains have not been considered in the results.

179 – Why is there a range for the ANI score when comparing only one isolate?

200-202 – what is the definition of common?

202 – is this phenotypic resistance or predicted resistance from genomic data?

209 – please see comments for 202.

211-212 – what is the definition of common. Please define the use of “common” throughout the manuscript.

255 – was resistance proven phenotypically or is this linked to other studies. If the latter, please site the relevant studies.

259 – please define the multiple resistance genes.

274 – Fig5 does not show information on strain country of origin. Therefore, this also needs to be linked back to the appropriate table as well to make this connection to the data, or this information overlayed on Fig. 5.

374 – the use of depicted here does not fit. Please change the wording.

380 – a final summary to the points being made in this section is missing.

381 – what is meant by sensation.

383 – what are these genes, are they cytolysin genes?

385 – is there a discussion as to why you think the strains from China are missing these genes and the suggested implications? What does this all mean?

386 – I suggest a new paragraph starting with the switch to Hyaluronidase.

Figures:

Fig 1. Please expand on the information provided in the legend. It is not clear which samples are depicted and in what order. It is not clear why some genes are annotated but others are not. What reference strain was used for this analysis? What are the different concentric rings? I would suggest breaking this figure up into A, B and C.

Fig 2. Please expand the information in the legend so that the figure is understandable on its own. What are the ref strains .

Figure 3. There is little information provided in the figure legend. Please expand the information in the legend so that the figure is understandable on its own. What do the different colours mean?

Figure 4. It should be “Multilocus sequence typing.” I would suggest that the “white box” is fig 4C.

---

## Round 0.2 · accepted · Accept

The authors have now addressed all of the reviewers previous comments, and the manuscript now appears ready for publication.

Reviewer 4 ·

Basic reporting

The authors have addressed the comments from reviewers and I recommend that the manuscript should be published.

Experimental design

The authors have addressed the comments from reviewers and I recommend that the manuscript should be published.

Validity of the findings

The authors have addressed the comments from reviewers and I recommend that the manuscript should be published.

Additional comments

The authors have addressed the comments from reviewers and I recommend that the manuscript should be published.